# Study on the impact of Kinesiophobia after Total Knee Arthroplasty on the rehabilitation of patients during hospitalization: A pilot study

**Yichao Yao** [ORCID]*, **Qi Zhang, Shaoning Cui, Xumeng Guo**

Department of Operating Room, Baoding First Central Hospital, Baoding, China

* 727307252@qq.com

**Data Availability Statement:** To protect patients privacy, the name of patients will not appeared in the data.Data is provided within the manuscript files. The datasets used and analyzed during the

## Abstract

The purpose of this study is to investigate the influence of kinesiophobia following Total Knee Arthroplasty (TKA) on the rehabilitation outcomes of patients during hospitalization, includes examining the trends in resting pain levels at various time points post-surgery, the trends in active flexion of the knee at various time points post-surgery, and the effects of kinesiophobia on the timing of first postoperative ambulation, the duration of postoperative hospital stay, and the results of the two-minute walk test on the day of discharge. Postoperative kinesiophobia in patients was identified using the Tampa Scale for Kinesiophobia (TSK), with 33 patients scoring >37 points and 35 patients scoring ≤37 points. Resting Pain levels were assessed using the Numerical Rating Scale (NRS) at various time points, including upon return to the ward (T1), the first (T2), second (T3), third (T4), fifth(T5) postoperative days, and the day of discharge (T6). Furthermore, active flexion of the knee joint was measured at different time points for both groups, including the first (T1*), third (T2*), and fifth (T3*) postoperative days, and the day of discharge (T4*). The first time out of bed activities of the two groups of patients were compared, along with the results of the 2-Minute Walk Test (2-MWT) on the day of discharge. The pain scores of patients in the kinesiophobia group at different time points after surgery were worse than those in the non-kinesiophobia group (P<0.05). There were significant time effects (F = 131.297;P<0.01), inter-group effects (F = 15.016; P<0.01), and interaction effects (F = 5.116; P<0.05). The active knee flexion of patients in the kinesiophobia group at different time points after surgery were worse than those in the non-kinesiophobia group (P<0.05).There were significant time effects (F = 628.258;P<0.01), inter-group effects (F = 16.546; P<0.01), and interaction effects (F = 66.025; P<0.01). Patients in the kinesiophobia group delayed getting out of bed for the first time (35.39±9.82vs28.77±9.81hours; P<0.01), had shorter activity time (4.12 ±1.36vs5.80±1.96minutes; P<0.01) and distance (19.12±3.36vs30.17±5.64meters; P<0.01), and experienced higher pain scores during activity (6.30±1.10vs5.48±0.95scoresP<0.05). Additionally, patients in the kinesiophobia group walked shorter distances in the 2-MWT test on the day of discharge compared to the non-kinesiophobia group (37.60 ±5.76vs50.68±5.37meters;P<0.05), and had longer hospitalization time (8.11±1.31vs50.68 ±5.37days; P<0.05). In short, compared to patients without kinesiophobia, the presence of

current study are available from the figshare (https://figshare.com). Doi:10.6084/m9.figshare.25422769".

**Funding:** The author(s) received no specific funding for this work.

**Competing interests:** The authors have declared that no competing interests exist.

kinesiophobia after TKA surgery significantly impacts the efficacy of early rehabilitation exercises during hospitalization. This impact is observed in pain perception, knee joint mobility, the 2-minute walk test, etc. Early identification of patients with kinesiophobia after TKA and timely intervention are necessary and beneficial.

## Introduction

Total knee arthroplasty (TKA) is considered the most effective treatment for the end-stage of knee osteoarthritis [1]. Postoperative rehabilitation exercises play a crucial role in enhancing joint function and mobility, as well as increasing patient satisfaction with the treatment [2]. Failure to properly perform rehabilitation exercises can significantly hinder the patient's recovery of knee joint function. Research findings indicate that approximately 30% of individuals who undergo knee replacement surgery experience adverse effects, including significant lower limb pain and dysfunction in the knee joint [3]. Kinesiophobia is a unique phenomenon where patients experience an exaggerated and irrational fear of physical activity or exercise due to prolonged chronic pain, resulting in heightened pain sensitivity [4]. Some studys pointed out that 20% of patients experience long-term chronic pain after knee replacement surgery [5]. A study found that a substantial proportion of knee replacement patients (55.8%) exhibited high levels of kinesiophobia at the time of hospital discharge [6]. The relationship between kinesiophobia and knee function is particularly concerning. Patients with high kinesiophobia are more likely to experience limitations in their range of motion and strength, which are critical for recovery after total knee arthroplasty (TKA). For example, kinesiophobia has been linked to lower knee flexion range of motion and decreased muscle strength, which are essential for performing daily activities and returning to pre-injury levels of function. Furthermore, kinesiophobia can exacerbate pain perception, as patients may interpret normal sensations as threatening, leading to increased pain and discomfort [6, 7]. Existing research has demonstrated the detrimental effects of kinesiophobia on long-term knee joint function recovery [8]. However, few studies have explored the impact of postoperative kinesiophobia on early rehabilitation exercises and early knee functional recovery after TKA. The existing studies on this topic are primarily conducted abroad, for example, a study conducted in Turkey highlighted that patients undergoing knee replacement surgery who experienced kinesiophobia exhibited significantly poorer exercise performance compared to those without kinesiophobia [9]. This was evident in measurements such as the 2-Minute Walk Test (2-MWT), pain assessment, knee range of motion, and other relevant indicators on the day of discharge. There is currently no such research in mainland China. In some countries, early rehabilitation exercises after TKA are supervised by physical therapists, potentially mitigating the negative effects of kinesiophobia on knee joint exercise rehabilitation [10]. However, in many parts of China, the scarcity of physical therapists and limited access to rehabilitation resources may result in a lack of professional guidance and supervision during early postoperative rehabilitation, potentially hindering the recovery process for patients with kinesiophobia [11]. The impact of kinesiophobia on early exercise levels and functional recovery after TKA remains unclear in current literature. The purpose of this study is to investigate the influence of kinesiophobia following Total Knee Arthroplasty(TKA) on the rehabilitation outcomes of patients during hospitalization, includes examining the trends in resting pain levels at various time points post-surgery, the trends in active flexion of the knee at various time points post-surgery, and the effects of kinesiophobia on the timing of first postoperative ambulation, the duration of postoperative hospital stay, and the results of the two-minute walk test on the day of discharge.

Based on the preceding discussion and existing research, we propose the following hypotheses. Hypothesis 1: At various measurement time points post-surgery, patients with kinesiophobia will exhibit higher resting pain scores than those without kinesiophobia. Hypothesis 2: At various measurement time points post-surgery, patients with kinesiophobia will demonstrate poorer knee joint flexion compared to those without kinesiophobia. Hypothesis 3: Compared to patients without kinesiophobia, those with kinesiophobia will experience a delayed time to ambulate for the first time after surgery, engage in shorter durations and distances of activity, and report higher pain scores during their initial mobilization. Furthermore, patients with kinesiophobia are expected to have longer postoperative hospitalization durations and perform worse on the two-minute walk test on the day of discharge.

## Methods and analysis

### Ethical approval

Ethical approval was obtained from the Ethics Committee of Baoding No.1 Central Hospital (2021–016).All study protocols were conformed to the principle of the Declaration of Helsinki. Each participant signed an informed consent form before entering the study.

### Subjects

The recruitment period started on 15 June 2021, and finished on 31 December 2021.Sixty-eight patients with knee osteoarthritis were conveniently recruited from a tertiary hospital in Baoding City, all recruits were patients undergoing unilateral knee replacement. Inclusion criteria for this study include meeting the diagnostic criteria for knee osteoarthritis, undergoing primary unilateral knee replacement surgery, being between the ages of 50–80 years old, having no cognitive impairment, providing informed consent, and being able to cooperate with the investigation. Exclusion criteria involve suffering from serious heart, lung, kidney, or other important organ diseases, having diseases of the nervous system or musculoskeletal system that affect movement, or refusing to participate in the investigation.

### Measures

The questionnaire was developed by the research team based on a review of the literature and input from experts. It includes demographic and disease-related information such as gender, age, education level, marital status, main caregiver, chronic diseases, affected limbs, years of pain, pain score, preoperative knee flexion, body mass index (BMI), and duration operation time. The Tampa Scale for Kinesiophobia (TSK) consists of 17 items scored on a 4-point Likert scale ranging from 1 to 4. A score above 37 indicates kinesiophobia. The Numerical Rating Scale (NRS) for pain assesses pain levels on a scale of 0–10, with 0 indicating no pain, 1–3 mild pain, 4–6 moderate pain, and 7–10 severe pain. Active flexion of the knee joint involves the patient lying supine with the lower limbs extended as the starting point. The knee joint is then actively flexed to its maximum angle, and a long-arm protractor is used to measure the angle between the line connecting the greater trochanter of the femur to the lateral femoral condyle and the line connecting the lateral femoral condyle to the lateral malleolus. Record the time when the patient first gets out of bed, including the interval between returning to the ward after surgery and the first time they get out of bed, activity duration, activity distance, and pain level during the activity. On the day of discharge, perform the 2-Minute Walk Test (2-MWT) with a walking aid and record the distance traveled. Lastly, record the postoperative hospitalization time as the number of days from the second day after surgery to the patient's discharge date.

## Research process

This study has been approved by the Ethics Committee of Baoding No.1 Central Hospital. Informed consent forms were signed by all participating patients. In the sampling process of this study, given the constraints of limited research resources and time, we employed a convenience sampling strategy. Specifically, during the study period, investigators assessed whether patients admitted to the hospital met the inclusion criteria by reviewing the medical record system. If a patient satisfied the inclusion criteria, the investigator would bring the informed consent form to the patient's bedside, provide a brief summary of the study's objectives, and explain the detailed research process. The patient would also be informed of their right to withdraw from the study at any time. Should the patient agree to participate, they would be required to sign the informed consent form. Conversely, if a patient declined to participate, they were assured that this decision would not negatively affect their treatment during hospitalization.

The patient's responsible nurse collected and filled out the general information questionnaire after the patient's admission to the hospital. Surgeries were conducted by a team of surgeons, with general anesthesia administered before the procedure. Adductor canal block was performed under ultrasound guidance. Prior to surgery, a pneumatic tourniquet was applied to the affected limb, with the pressure set between 40 kPa and 45 kPa. All surgeries utilized the medial approach to the patella, completed osteotomy, and employed Biomet Vaguard PS series prostheses. No drainage tube was left postoperatively.

Following the operation, a rehabilitation exercise instruction manual will be provided to the patient by the nurse, while the attending doctor will guide them on the exercise techniques. The manual outlines daily postoperative rehabilitation exercises such as ankle pump exercises, quadriceps isometric exercises, passive and active knee flexion exercises, and specifies the frequency and duration of these exercises. Exercise measures include both activities in bed and out of bed. For bed exercises, patients are advised to start moving their toes and perform ankle pump exercises on the day after surgery. Ankle pump exercises should last for 5 minutes each time, once an hour. On the second day post-surgery, patients should begin isometric contraction exercises for the quadriceps and hamstrings, holding for 10 seconds each time, in sets of 10 repetitions, twice a day. Straight leg raising training and active knee flexion training can be initiated on the third postoperative day. For straight leg raise training, patients should aim for 20 repetitions per set, with 4 sets per day. Active knee flexion training involves actively bending the knees as much as tolerated, 10 times per set, for 4 sets per day. These bed exercises should continue until the day of discharge. Patients are encouraged to start ambulation the following day. When getting out of bed, patients should use a walking aid, walking the maximum distance they can tolerate, twice a day. Due to the lack of physical therapist supervision and limited medical resources, the patient was initially guided and supervised by the responsible nurse during his first attempt to get out of bed. Subsequent rehabilitation exercises were carried out by the patient with assistance from his family members.

Prior to the questionnaire survey, we conducted a unified training session for the two team members responsible for the assessment. This training ensured that they were familiar with the content of the questionnaire and the survey process, enabling them to accurately address any questions posed by the patients. During the investigation, the evaluators were instructed to use standardized guidelines to explain the study's purpose and the procedure for completing the questionnaire to the patients. The day after the patient regained consciousness and returned to the ward, the research team distributed the TSK rating scale to the patient, who completed it on site. In cases where the patient had difficulty a understanding certain items, the research team members clarified them before the patient filled them out. After the patient completes the questionnaire, the evaluator should conduct a preliminary review of the

responses on-site to identify any missing items, obvious errors, or illogical answers, and confirm these with the respondent promptly. Individuals who score greater than 37 points on the questionnaire are classified as having kinesiophobia, while those who score 37 points or less are classified as not having kinesiophobia. The patient's resting pain level was evaluated by evaluator on multiple days postoperatively, including upon returning to the ward after surgery (T1), the first postoperative day (T2), the second postoperative day (T3), the third postoperative day (T4), the fifth postoperative day (T5), and the day of discharge (T6). To ensure accurate pain assessment, the evaluator should provide consistent instructions to explain the Numerical Rating Scale to the patient prior to each evaluation. The assessment should be conducted while the patient is conscious, and the evaluator should confirm the findings with the patient following the assessment. When the patient first got out of bed, they were accompanied by the responsible nurse, a research team member, and a family member. The research team recorded the time of activity initiation, duration, distance covered, and pain level during the activity. The patient's knee joint active flexion was measured by the attending physician on the first postoperative day (T1*), third postoperative day (T2*), fifth postoperative day (T3*), and the day of discharge (T4*) during ward rounds. To ensure the accuracy of the measurements, each assessment is conducted by the attending physician, who employs standardized instructions to explain the measurement method to the patient. Additionally, the same ruler is utilized to measure the patient's active knee flexion. Additionally, a 2-MWT was conducted on the morning of discharge under the supervision of the attending doctor, responsible nurse, a research team member, and a family member. The test was performed with the assistance of a walker, and the patient was instructed to complete it as quickly as tolerated. The postoperative hospitalization time was recorded by the research team members based on the patient's medical records.

## Sample size

Sample size calculation was performed using power analysis and sample size (PASS) software (version 2021).The pain scores of the research object were used as the observed outcome indicators. The value of alpha was set as 0.05, power was set as 0.9, number of repeated measurements set as 6, and according to previous literature [9], means was set as 3, standard deviation was set as 3, conditional correlation coefficient set as 0.5.The result showed that one group was expected to recruit 31 patients.

## Statistical analysis

SPSS 21.0 statistical software(IBM SPSS Statistics 21) and GraphPad were utilized for data processing and statistical analysis. Group comparisons of count data were conducted using the chi-square or Fisher exact test and presented as numbers and percentages. For all data sets normality of distribution was verified with the Shapiro–Wilk test, with Levene's test to confirm homogeneity of variances. For normally distributed measurement data, such as resting pain, active flexion of knee joint,2-MWT results et al, the mean ± standard deviation was expressed, with group comparisons made using two independent sample t-test. Data with repeated measures, such as changes in postoperative pain level and active flexion of the knee joint, were analyzed with repeated-measures analysis of variance. A p-value $<0.05$ was considered statistically significant for all tests.

## Results

### Comparison of general data of the two groups

A total of 68 patients were included in this study. Of these, 33 patients were classified in the kinesiophobia group (TSK>37 points) and 35 patients in the non-kinesiophobia group

(TSK≤37 points). Upon comparing the general information of the two patient groups, no statistical differences were observed. Refer to Table 1 for detailed results.

## Comparison of resting pain scores between the two groups at different time points after surgery

The knee joint resting pain scores of the two groups of patients at different time points after surgery were compared using repeated measures analysis of variance. Both sets of data met the criteria for normality (Shapiro–Wilk test) and homogeneity of variances(Levene's test). The Mauchly test of sphericity indicated that the variances of differences between groups were not equal(W = 0.196, P<0.01). Hence, the results of the multivariate test were considered. The analysis revealed a statistically significant time effect (F = 131.297, p<0.01), indicating a decreasing trend in postoperative knee pain for both patient groups over time. Furthermore, the interaction between the time factor and grouping was also statistically significant (F = 5.116, P<0.01), suggesting that the presence of kinesiophobia influenced the trajectory of pain intensity reduction over time in both groups. Fig 1 illustrates that the decline in postoperative pain scores for patients in the kinesiophobia group was less pronounced compared to those in the non-kinesiophobia group. Additionally, the inter-group effect was found to be statistically significant (F = 15.016, P<0.01), highlighting a significant difference in pain scores between the two groups at each time point. Detailed results can be found in Table 2.

## Comparison of active knee flexion between the two groups at different time points after surgery

The comparison of postoperative active knee flexion between two groups of patients at different time points was conducted using repeated measures analysis of variance. Both sets of data met the criteria for normality (Shapiro–Wilk test) and homogeneity of variances (Levene's test). The Mauchly test of sphericity indicated that the variances of differences between groups were not equal(W = 0.239, P<0.01). Hence, the results of the multivariate test were considered. The analysis revealed that the time factor had a statistically significant effect (F = 428.297, p<0.01), indicating that active knee flexion tended to increase over time for both patient groups. Additionally, the interaction between the time factor and grouping was statistically significant (F = 16.549, p < 0.01), suggesting that the presence of kinesiophobia influenced the rate of knee flexion improvement over time in both groups. Fig 2 illustrates that patients in the kinesiophobia group exhibited a weaker increase in active knee flexion compared to those in the non-kinesiophobia group postoperatively. The between-group difference was statistically significant (F = 66.025, p<0.01), indicating a significant variation in active knee flexion between the two patient groups at each time point. Specific results can be found in Table 3.

## Comparison of the first out-of-bed activity, postoperative hospitalization days, and 2-MWT results on the day of discharge between the two groups

When comparing the initial activities of patients in the kinesiophobia group with those in the non-kinesiophobia group, it was observed that patients in the kinesiophobia group took longer to get out of bed after surgery. Additionally, they had shorter durations for their first time getting out of bed and reported higher pain scores during this activity. These differences were found to be statistically significant. Furthermore, patients in the kinesiophobia group had longer postoperative hospitalization days compared to those in the non-kinesiophobia group, with a statistically significant variance. On the day of discharge, patients in the kinesiophobia

**Table 1. Comparison of general data of the two groups (n = 68).**

| | TSK≤37 | TSK>37 | $\chi^2/t$ | *P*-value |
|---|---|---|---|---|
| | n = 35 | n = 33 | | |
| **Gender** | | | 0.084 | >0.05 |
| Male | 15 | 13 | | |
| Female | 20 | 20 | | |
| **Age(year)** | | | -1.212 | >0.05 |
| | 64.05±2.62 | 64.72±1.87 | | |
| **Education lever** | | | 1.975 | >0.05 |
| Primary school | 15 | 16 | | |
| Junior school | 10 | 12 | | |
| Senior high school | 7 | 4 | | |
| College degree or above | 3 | 1 | | |
| **Marriage** | | | 0.101 | >0.05 |
| Married | 31 | 30 | | |
| Else | 4 | 3 | | |
| **Nationality** | | | 0.006 | >0.05 |
| The Han nationality | 32 | 31 | | |
| Else | 3 | 2 | | |
| **Primary caregiver** | | | 1.853 | >0.05 |
| Spouse | 24 | 25 | | |
| Offspring | 11 | 7 | | |
| Others | 0 | 1 | | |
| **Other chronic diseases** | | | 0.197 | >0.05 |
| None | 5 | 6 | | |
| 1–2 | 17 | 15 | | |
| 3 or more | 13 | 12 | | |
| **Surgical limb** | | | 0.003 | >0.05 |
| Left knee | 16 | 15 | | |
| Right knee | 19 | 18 | | |
| **Duration of knee pain** | | | 0.502 | >0.05 |
| <1year | 6 | 5 | | |
| 1year—3year | 21 | 19 | | |
| >3year | 8 | 9 | | |
| **Preoperative pain score** | | | 0.848 | >0.05 |
| | 3.85±1.03 | 3.63±1.11 | | |
| **Preoperative knee flexion** | | | 1.092 | >0.05 |
| | 90.15±5.21 | 91.22±4.35 | | |
| **BMI(kg/m²)** | | | -1.701 | >0.05 |
| | 23.60±1.28 | 24.63±3.25 | | |
| **Duration of the operation (h)** | | | -0.635 | >0.05 |
| | 2.52±1.24 | 2.58±1.37 | | |

Data are presented as mean±standard deviation unless otherwise indicated. BMI, body mass index.

group walked shorter distances in the 2-Minute Walk Test (2-MWT) in comparison to the non-kinesiophobia group, with statistical significance. Detailed results can be found in Table 4.

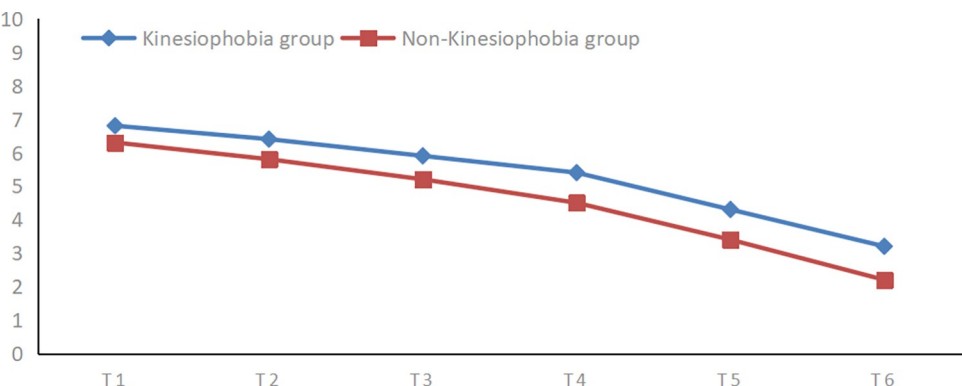

**Fig 1. The change trend of postoperative resting pain in two groups.** The blue line represents the change trend of pain scores in the kinesiophobia group. The red line represents the change trend of pain scores in the non-kinesiophobia. The downward trend in postoperative pain scores in the kinesiophobia group was not as obvious as that in the non-kinesiophobia group.T1:upon return to ward after ssurgery,T2:the first postoperative day,T3:the second postoperative day,T4:the third postoperative day,T5:the fifth postoperative day,T6: the day of discharge.

## Discussion

The fear-avoidance model is a psychological framework that explains how fear of pain can lead to avoidance behaviors, which in turn can significantly impact a patient's psychological state and recovery process. This model posits that when individuals experience pain, they may develop negative beliefs about physical activity and their ability to cope with pain, leading to a cycle of fear and avoidance [12]. Postoperative pain significantly influences early rehabilitation exercises in patients following knee replacement surgery [8]. This study revealed that patients in the kinesiophobia group experienced higher pain scores compared to those in the non-kinesiophbia at multiple time points (T1, T2, T3, T4, T5, and T6), with statistically significant differences. Hande's study also highlighted that on the day of discharge, the group exhibiting kinesiophobia reported higher pain scores compared to the group without kinesiophobia [9]. Our study's findings align with this observation. Demonstrated that nearly 25% of patients with a knee replacement suffer from chronic pain [12], these psychological and biological alterations need to be considered by clinicians and physical therapist when planning treatment. Pain perception is subjective and influenced not only by the underlying disease but also by the individual's psychological state, which is a crucial factor to consider [13]. The psychological state of patients affected by the fear-avoidance model can deteriorate as a result of these avoidance behaviors. Patients may experience increased anxiety, depression, and a sense of helplessness due to their inability to engage in normal activities. This emotional distress can

**Table 2. Repeated measures analysis of variance results of resting pain score at different time points between the two groups after surgery.**

|  | T1 | T2 | T3 | T4 | T5 | T6 | Time effect | Inter-group effect | Interaction effect |
|---|---|---|---|---|---|---|---|---|---|
| **Kinesiophobia group** | 6.81±0.91 | 6.21±0.99 | 6.03±0.68 | 5.24±1.03 | 4.3±1.13 | 3.15±0.79 | 131.297$^{\triangle\triangle}$ | 15.016$^{\triangle\triangle}$ | 5.116$^{\triangle}$ |
| **Non-Kinesiophobia group** | 6.05±1.25 | 5.20±0.63 | 5.20±0.63 | 4.51±0.85 | 3.25±1.03 | 2.40±0.73 |  |  |  |
| ***t*-value** | 2.835 | 2.820 | 5.201 | 3.179 | 3.974 | 4.048 |  |  |  |
| ***p*-value** | <0.01 | <0.01 | <0.01 | <0.01 | <0.01 | <0.01 |  |  |  |

Data are presented as mean±standard deviation unless otherwise indicated. $\triangle$p<0.01 statistically significant difference. $\triangle\triangle$p<0.001 statistically significant difference. T1:upon return to ward after surgery,T2:the first postoperative day,T3:the second postoperative day,T4:the third postoperative day,T5:the fifth postoperative day,T6: the day of discharge.

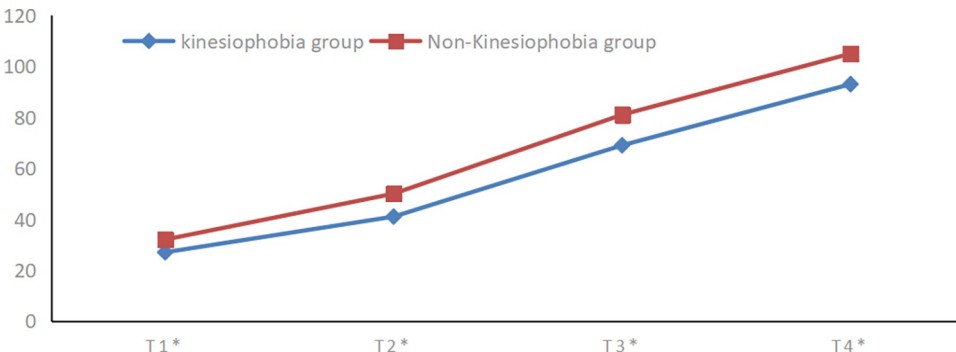

**Fig 2. The change trend of postoperative active knee flexion in two groups.** The blue line represents the change trend of active knee flexion in the kinesiophobia group. The red line represents the change trend of active knee flexion in the non-kinesiophobia. The upward trend in postoperative active knee flexion in the kinesiophobia group was not as obvious as that in the non-kinesiophobia group.T1*:the first postoperative day,T2*:the third postoperative day,T3*:the fifth postoperative day,T4*: the day of discharge.

further exacerbate their perception of pain and disability, creating a vicious cycle where fear leads to avoidance, which in turn leads to increased pain and disability 4. The model highlights that fear-avoidance beliefs can lead to chronic pain conditions, as the lack of movement and activity can result in physical deconditioning and a reduced ability to cope with pain [14]. Factors such as prosthetic implantation, bone and soft tissue damage, joint swelling, and the use of tourniquets during knee replacement surgery can contribute to postoperative pain in patients [15, 16]. Patients with kinesiophobia, however, tend to be more sensitive to pain and may experience heightened pain perception. Especially since 60% of the increase in pain score could be related to the presence of kinesiophobia [17]. Furthermore, patients suffering from kinesiophobia may, in order to prevent a future injury, develop hypervigilance. This state has been identified as a crucial contributor to the development of chronic pain [18] psychosocial factors are known to influence the biological processes that are crucial for recovery. For instance, neuroplastic changes in central nervous system regions associated with the processing of sensory information, emotions and pain are more likely to occur in patients with kinesiophobia. These changes interfere with the normal neurophysiological modulation of pain, and will in turn cause more pain-awareness and disability [19]. Additionally, these patients often exhibit poor pain coping strategies and low self-efficacy [20], potentially explaining why they report higher pain scores than those without kinesiophobia. Previous research finding has demonstrated a positive correlation between patients' pain scores and levels of kinesiophobia, indicating that higher TSK scores are associated with lower pain thresholds [9]. The findings of this study further support this relationship.

**Table 3. Repeated measures analysis of variance results of active knee flexion at different time points between the two groups after surgery.**

|  | T1* | T2* | T3* | T4* | Time effect | Inter-group effect | Interaction effect |
|---|---|---|---|---|---|---|---|
| **Kinesiophobia group** | 27.18±2.81 | 41.75±3.88 | 69.15±5.47 | 93.66±3.09 | 628.258△△ | 16.546△△ | 66.025△△ |
| **Non-Kinesiophobia group** | 32.34±4.15 | 50.20±4.49 | 80.85±4.17 | 105.20±10.61 |  |  |  |
| *t*-value | -5.966 | -8.272 | -9.947 | -6.001 |  |  |  |
| *p*-value | <0.05 | <0.05 | <0.05 | <0.05 |  |  |  |

Data are presented as mean±standard deviation unless otherwise indicated. △p<0.01 statistically significant difference. △△p<0.001 statistically significant difference. T1*:the first postoperative day,T2*:the third postoperative day,T3*:the fifth postoperative day,T4*: the day of discharge.

**Table 4. Comparison of the first out-of-bed activity, postoperative hospitalization days, and 2-MWT results on the day of discharge between the two groups.**

|  | First time to get out of bed (h) | Duration of first activity (min) | Distance of first activity (m) | Pain score of first activity | Postoperative hospital stay (d) | 2-MWT on day discharge (m) |
|---|---|---|---|---|---|---|
| **Kinesiophobia group** | 35.39±9.82 | 4.12±1.36 | 19.12±3.36 | 6.30±1.10 | 8.11±1.31 | 37.60±5.76 |
| **Non-Kinesiophobia group** | 28.77±9.81 | 5.80±1.96 | 30.17±5.64 | 5.48±0.95 | 7.20±0.90 | 50.68±5.37 |
| *t*-value | 2.779 | -4.006 | 9.735 | -3.278 | 4.225 | 9.678 |
| *p*-value | <0.01 | <0.01 | <0.01 | <0.05 | <0.01 | <0.05 |

Data are presented as mean±standard deviation unless otherwise indicated.2-MWT: 2-Minute Walk Test.

The findings of this study indicate that patients in the kinesiophobia group exhibited significantly smaller active flexion of the knee joint at T1*, T2*, T3*, and T4* time points compared to those in the non-kinesiophobia group. Specifically, patients require a minimum of 67˚ of knee flexion for walking on level ground, 90˚ for ascending stairs, 83˚ for descending stairs, and 115˚ for standing up from a sofa [8].This highlights the importance of knee flexion as a key prognostic indicator for functional recovery and satisfaction after TKA surgery in patients with osteoarthritis. Some scholars have noted that an increase of one point in the TSK score of patients following knee replacement surgery is associated with a decrease in knee flexion of approximately 1/2-2/3 degrees [21]. Apart from surgical factors, factors such as BMI, preoperative knee joint mobility, and postoperative active exercise also play a role in determining knee flexion after surgery [22]. Patients with kinesiophobia tend to avoid and fear exercise due to their unique psychological characteristics. Research has indicated that 47% of the variability in active knee flexion can be attributed to the level of kinesiophobia, regardless of factors such as pain, age, BMI, and gender [9]. Effective physical therapy interventions have the potential to mitigate the negative effects of kinesiophobia on rehabilitation exercises, leading to pain reduction, increased muscle strength, and decreased levels of kinesiophobia [10]. However, the involvement of a skilled physical therapist is essential for achieving these outcomes. A personalized exercise and kinesiophobia management plan developed by professional physical therapists can significantly enhance knee joint function, including knee flexion, and overall quality of life in patients with kinesiophobia after TKA surgery. This approach can result in a recovery level comparable to that of patients without kinesiophobia [2]. In this study, medical staff conveyed postoperative exercise methods to patients and emphasized the importance of early exercise. However, due to human resource constraints, it was challenging to monitor the frequency and intensity of patients' daily exercises. As a result, evaluating and ensuring the quality of their exercise completion posed difficulties. The current shortage of medical staff and rehabilitation resources leads to a lack of supervision over patients' spontaneous exercise behavior, which often results in poor patient exercise compliance. Patients may fail to adhere to the prescribed items, frequency, and intensity of the rehabilitation plan, or may stop exercising on their own accord.

The results of the 2-MWT on the day of discharge indicated that patients in the kinesiophobia group had a shorter walking distance compared to those in the non-kinesiophobia group. Previous research demonstrated that higher levels of kinesiophobia in patients were associated with shorter distances covered in the 2-MWT, and 41% of the variation in the 2-MWT can be explained by the level of kinesiophobia [9]. Our study is consistent with previous studies. Apart from factors like lower limb muscle strength and joint function, the motivation of patients to exercise also played a significant role in the results of the walking test. Patients with kinesiophobia often have a tendency to avoid movement due to misconceptions such as fear of pain or exacerbating injuries, leading to reduced exercise motivation [23]. This reluctance to

engage in physical activity can impact the effectiveness of rehabilitation exercises, resulting in poor recovery of quadriceps muscle strength and knee joint function, ultimately affecting the 2-MWT outcomes on the day of discharge.

The discharge criteria in this study focused on successful incision healing, achieving 90° of active knee flexion, and absence of serious postoperative complications. Patients in the kinesiophobia group had slightly longer hospitalization times compared to those in the non-kinesiophobia group. Patients in the non-kinesiophobia group initiated exercise earlier and experienced better exercise outcomes, which facilitated incision healing, reduced postoperative complications, and enhanced postoperative muscle strength and knee joint function. Consequently, their recovery outperformed that of patients in the kinesiophobia group, allowing them to meet discharge criteria sooner.

Assessing the degree of kinesiophobia in patients after knee replacement surgery is crucial for developing personalized intervention plans that can enhance recovery outcomes. Conducting structured interviews can help clinicians understand the patient's experiences and fears related to movement. Open-ended questions about their concerns regarding pain during rehabilitation activities can reveal underlying kinesiophobia [24]. Providing patients with clear, consistent information about the recovery process and the importance of movement can help alleviate fears. Educational interventions should focus on the benefits of early mobilization and the expected pain levels during rehabilitation [7]. Encouraging the involvement of family and friends in the rehabilitation process can provide emotional support and reduce feelings of isolation. Research indicates that social support plays a crucial role in recovery, as patients who feel supported are more likely to engage in rehabilitation activities and adhere to exercise regimens [25]. A comprehensive, multidisciplinary approach that includes physiotherapists, psychologists, and occupational therapists can enhance recovery. This team can collaboratively address both the physical and psychological aspects of rehabilitation, ensuring that patients receive holistic care that targets kinesiophobia alongside physical recovery [26]. In short, early identification of patients with kinesiophobia after TKA and timely intervention are necessary and beneficial.

The study's limitations include the use of a sample from a single source and the inability to test the patient's muscle strength due to restrictions in research conditions. Due to a shortage of physical therapists and medical staff, not all patients can exercise under professional guidance and supervision. After initial guidance, subsequent exercises rely on the patient's self-initiative. This makes it challenging to measure the completion rate and quality of exercise, highlighting the common scenario in China regarding early post-knee replacement surgery rehabilitation.

## Conclusion

In short, compared to patients without kinesiophobia, the presence of kinesiophobia after TKA surgery significantly impacts the efficacy of early rehabilitation exercises during hospitalization. This impact is observed in pain perception, knee joint mobility, the 2-minute walk test, etc. Early identification of patients with kinesiophobia after TKA and timely intervention are necessary and beneficial.

## Supporting information

**S1 Dataset.**
(XLSX)

## Author Contributions

**Investigation:** Qi Zhang, Shaoning Cui.

**Methodology:** Qi Zhang, Shaoning Cui, Xumeng Guo.

**Resources:** Xumeng Guo.

**Supervision:** Xumeng Guo.

**Writing – original draft:** Yichao Yao.

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
