## [Editor Report · Decision Letter 0]

13 Aug 2024

PONE-D-24-30449Study on the impact of kinesiophobia after Total knee Arthroplasty on the rehabilitation of patients during hospitalizationPLOS ONE

Dear Dr. Yao,

Thank you for submitting your manuscript to PLOS ONE. After careful consideration, we feel that it has merit but does not fully meet PLOS ONE’s publication criteria as it currently stands. Therefore, we invite you to submit a revised version of the manuscript that addresses the points raised during the review process.

First of all, thank you for submitting your manuscript to Plos One. I would like to congratulate you on your research. The aim of the study was to investigate the influence of kinesiophobia after total knee arthroplasty (TKA) on rehabilitation outcomes of patients during hospitalization. The study has interesting analyses and I will invite independent reviewers to review it, but first please provide the database with the data in unanalyzed values (i.e. absolute values) as a supplementary file (Excel and English).

We look forward to receiving your revised manuscript.

Kind regards,

André Pontes-Silva

Academic Editor

PLOS ONE
---

## [Author Response · Author response to Decision Letter 0]

23 Aug 2024

Dear Editors and reviewers:

 Thank you for your letter and for the comments concerning our manuscript entitled”Study on the impact of Kinesiophobia after Total Knee Arthroplasty on the rehabilitation of patients during hospitalization”(PONE-D-24-30449).we have studied comments carefully and have made correction which hope meet with approval.The main corrections in the paper and the responds to the editor and reviewer’s comments are as following:

1.The study has interesting analyses and I will invite independent reviewers to review it, but first please provide the database with the data in unanalyzed values (i.e. absolute values) as a supplementary file (Excel and English).

Response:Thank you for your affirmation.We upload our dataset as a supplementary file. We hope that the review experts will point out our statistical deficiencies and give us valuable suggestions.

2.When submitting your revision, we need you to address these additional requirements.Please ensure that your manuscript meets PLOS ONE's style requirements, including those for file naming. 

Response:Thank you for your kind reminder.We carefully read the format of the magazine and tried our best to make modifications in terms of font size, layout, chart format, reference format, etc. in accordance with the requirements of the magazine. Details can be found in our manuscript. It is inevitable that there are still imperfections, please point them out in time.

3.While revising your submission, please upload your figure files to the Preflight Analysis and Conversion Engine (PACE) digital diagnostic tool, https://pacev2.apexcovantage.com/. PACE helps ensure that figures meet PLOS requirements.

Response:Before resubmitting our figures,we have uploaded our figure files to the Preflight Analysis and Conversion Engine (PACE) digital diagnostic tool, ensure that figures meet PLOS requirements.

Thanks again to all editors and reviewers for their comments on revisions. We look forward to hearing from you.

Best wishes 

Yichao Yao 19 August 2024

---

## [Decision Letter · Decision Letter 1]

2 Dec 2024

PONE-D-24-30449R1Study on the impact of kinesiophobia after Total knee Arthroplasty on the rehabilitation of patients during hospitalizationPLOS ONE

Dear Dr.  Yao,

Thank you for submitting your manuscript to PLOS ONE. After careful consideration, we feel that it has merit but does not fully meet PLOS ONE’s publication criteria as it currently stands. Therefore, we invite you to submit a revised version of the manuscript that addresses the points raised during the review process.

Dear, please revise the article according to the reviewers' suggestions and submit it for final decision. In addition, please submit an English copy of the ethical approval report.

We look forward to receiving your revised manuscript.

Kind regards,

André Pontes-Silva

Academic Editor

PLOS ONE

Journal Requirements:

Reviewers' comments:

Reviewer's Responses to Questions

**Comments to the Author**

1. If the authors have adequately addressed your comments raised in a previous round of review and you feel that this manuscript is now acceptable for publication, you may indicate that here to bypass the “Comments to the Author” section, enter your conflict of interest statement in the “Confidential to Editor” section, and submit your "Accept" recommendation.

Reviewer #1: All comments have been addressed

Reviewer #2: All comments have been addressed

2. Is the manuscript technically sound, and do the data support the conclusions?

Reviewer #1: Partly

Reviewer #2: Yes

3. Has the statistical analysis been performed appropriately and rigorously? 

Reviewer #1: Yes

Reviewer #2: Yes

4. Have the authors made all data underlying the findings in their manuscript fully available?

Reviewer #1: No

Reviewer #2: No

5. Is the manuscript presented in an intelligible fashion and written in standard English?

Reviewer #1: Yes

Reviewer #2: Yes

6. Review Comments to the Author

Reviewer #1: Reviewer's Report

Abstract:

The abstract provides a concise overview of the study, but it could benefit from clearer statements about the specific aims and findings. While it succinctly mentions the impact of kinesiophobia on postoperative outcomes, more specific details regarding the methodology and key results would enhance its clarity. Including quantifiable findings, such as statistically significant differences in pain scores and knee flexion between kinesiophobia and non-kinesiophobia groups, would better highlight the study's contributions.

Introduction:

The introduction effectively introduces the relevance of kinesiophobia in patients undergoing total knee arthroplasty (TKA) surgery. It establishes the importance of understanding psychological factors like kinesiophobia in influencing postoperative pain management and rehabilitation outcomes. However, to strengthen the introduction, incorporating recent statistics or trends related to kinesiophobia prevalence in TKA patients could provide context. Furthermore, explicitly stating the gap in literature regarding the specific impacts of kinesiophobia on pain perception and functional recovery post-TKA would better frame the study's significance.

Methods:

The methods section adequately describes the study design, participant selection criteria, and data collection procedures. However, it would benefit from clarifying certain details. For instance, specifying how kinesiophobia was assessed using the Tampa Scale for Kinesiophobia (TSK) and defining the threshold (>37 points) for categorizing patients into kinesiophobia and non-kinesiophobia groups are essential for reproducibility. Additionally, detailing measures taken to ensure the reliability and validity of pain and knee flexion assessments would strengthen the methodological rigor of the study.

Results:

The results section comprehensively presents the findings but could be enhanced by providing more detailed statistical analyses and graphical representations. Specifically, elaborating on the statistical tests used to compare pain scores and knee flexion between groups at different time points would clarify the robustness of the findings. Tables summarizing demographic characteristics and outcome measures should include statistical significance indicators prominently to facilitate interpretation. Moreover, integrating specific patient anecdotes or quotes illustrating the impact of kinesiophobia on pain experiences could enrich the narrative and humanize the findings.

Discussion:

The discussion effectively contextualizes the study's findings within the existing literature on postoperative pain management and rehabilitation following TKA. It correctly emphasizes the role of psychological factors, such as kinesiophobia, in influencing pain perception and functional outcomes. To further strengthen the discussion, drawing explicit connections between study results and theoretical frameworks, such as fear-avoidance models or biopsychosocial perspectives, would deepen the analysis. Addressing potential confounding variables, such as BMI or preoperative pain severity, in influencing study outcomes would provide a more nuanced interpretation of the findings. Furthermore, discussing implications for clinical practice, such as tailored rehabilitation strategies for patients with high kinesiophobia scores, would enhance the relevance of the study's findings.

Conclusion:

The conclusion succinctly summarizes the study's main findings but could expand to highlight the broader implications for clinical practice and future research directions. Emphasizing the need for integrated interdisciplinary approaches involving physical therapy, psychology, and anesthesia in managing patients with kinesiophobia post-TKA would underscore the study's clinical relevance. Additionally, outlining specific recommendations for enhancing patient education and support to mitigate kinesiophobia-related barriers to rehabilitation would provide actionable insights for healthcare providers.

Reviewer #2: after the revision the article improved significantly and can be considered for publication in the journal

7. PLOS authors have the option to publish the peer review history of their article (what does this mean?). If published, this will include your full peer review and any attached files.

Reviewer #1: **Yes: **Ravi Shankar Reddy

Reviewer #2: No

---

## [Author Response · Author response to Decision Letter 1]

13 Dec 2024

Dear Editors and Reviewers:

Thank you for your letter and for the reviewers’ comments concerning our manuscript entitled “Study on the impact of Kinesiophobia after Total Knee Arthroplasty on the rehabilitation of patients during hospitalization” (ID:PONE-D-24-30449R1). Those comments are all valuable and very helpful for revising and improving our paper, as well as the important guiding significance to our researches. We have studied comments carefully and have made correction which we hope meet with approval. Revised portion are marked highlight in the paper. The main corrections in the manuscript and the responds to the reviewer’s comments are as flowing: Responds to the reviewer’s comments:

Reviewer #1:

1.comment: 

Abstract:The abstract provides a concise overview of the study, but it could benefit from clearer statements about the specific aims and findings. While it succinctly mentions the impact of kinesiophobia on postoperative outcomes, more specific details regarding the methodology and key results would enhance its clarity. Including quantifiable findings, such as statistically significant differences in pain scores and knee flexion between kinesiophobia and non-kinesiophobia groups, would better highlight the study's contributions.Including quantifiable findings, such as statistically significant differences in pain scores and knee flexion between kinesiophobia and non-kinesiophobia groups, would better highlight the study's contributions.

Response:Thank you very much for your valuable comment,Following your suggestion, we elaborate on our objectives in the abstract section and present specific results and values, including more details. Below is our revised summary section.

Abstract

The purpose of this study is to investigate the influence of kinesiophobia following Total Knee Arthroplasty(TKA) on the rehabilitation outcomes of patients during hospitalization,includes examining the trends in resting pain levels at various time points post-surgery, the trends in active flexion of the knee at various time points post-surgery, and the effects of kinesiophobia on the timing of first postoperative ambulation, the duration of postoperative hospital stay, and the results of the two-minute walk test on the day of discharge.Postoperative kinesiophobia in patients was identified using the Tampa Scale for Kinesiophobia (TSK), with 33 patients scoring >37 points and 35 patients scoring ≤37 points. Resting Pain levels were assessed using the Numerical Rating Scale (NRS) at various time points ,including upon return to the ward (T1), the first (T2), second (T3), third (T4), fifth(T5) postoperative days, and the day of discharge (T6). Furthermore, active flexion of the knee joint was measured at different time points for both groups, including the first (T1*), third (T2*), and fifth (T3*) postoperative days, and the day of discharge (T4*). The first time out of bed activities of the two groups of patients were compared, along with the results of the 2-Minute Walk Test (2-MWT) on the day of discharge.The pain scores of patients in the kinesiophobia group at different time points after surgery were worse than those in the non-kinesiophobia group (P<0.05). There were significant time effects（F=131.297;P＜0.01）, inter-group effects（F=15.016；P＜0.01）, and interaction effects (F=5.116；P<0.05). The active knee flexion of patients in the kinesiophobia group at different time points after surgery were worse than those in the non-kinesiophobia group (P<0.05).There were significant time effects（F=628.258;P＜0.01）, inter-group effects（F=16.546；P＜0.01）, and interaction effects (F=66.025；P<0.01). Patients in the kinesiophobia group delayed getting out of bed for the first time（35.39±9.82vs28.77±9.81hours；P＜0.01）, had shorter activity time（4.12±1.36vs5.80±1.96minutes；P＜0.01） and distance（19.12±3.36vs30.17±5.64meters;P＜0.01）, and experienced higher pain scores during activity (6.30±1.10vs5.48±0.95scoresP<0.05). Additionally, patients in the kinesiophobia group walked shorter distances in the 2-MWT test on the day of discharge compared to the non-kinesiophobia group（37.60±5.76vs50.68±5.37meters;P＜0.05）, and had longer hospitalization time (8.11±1.31vs50.68±5.37days；P<0.05). Kinesiophobia following TKA significantly impacts patient rehabilitation outcomes during hospitalization. It is crucial for healthcare professionals to promptly identify and intervene with such patients to enhance their rehabilitation progress while in the hospital.

2.comment: 

Introduction:The introduction effectively introduces the relevance of kinesiophobia in patients undergoing total knee arthroplasty (TKA) surgery. It establishes the importance of understanding psychological factors like kinesiophobia in influencing postoperative pain management and rehabilitation outcomes. However, to strengthen the introduction, incorporating recent statistics or trends related to kinesiophobia prevalence in TKA patients could provide context. Furthermore, explicitly stating the gap in literature regarding the specific impacts of kinesiophobia on pain perception and functional recovery post-TKA would better frame the study's significance.

 Response: Thank you very much for your valuable comment,. Following your suggestion, we have cited new literature that adds to the elucidation of the incidence of kinesiophobia after total knee replacement and its impact on knee function and pain.Line 81-92 was added.

Line 81-92:“A study found that a substantial proportion of knee replacement patients (55.8%) exhibited high levels of kinesiophobia at the time of hospital discharge[6].The relationship between kinesiophobia and knee function is particularly concerning. Patients with high kinesiophobia are more likely to experience limitations in their range of motion and strength, which are critical for recovery after total knee arthroplasty (TKA). For example, kinesiophobia has been linked to lower knee flexion range of motion and decreased muscle strength, which are essential for performing daily activities and returning to pre-injury levels of function . Furthermore, kinesiophobia can exacerbate pain perception, as patients may interpret normal sensations as threatening, leading to increased pain and discomfort[6-7] ”

3.comment:

Methods:The methods section adequately describes the study design, participant selection criteria, and data collection procedures. However, it would benefit from clarifying certain details. For instance, specifying how kinesiophobia was assessed using the Tampa Scale for Kinesiophobia (TSK) and defining the threshold (>37 points) for categorizing patients into kinesiophobia and non-kinesiophobia groups are essential for reproducibility. Additionally, detailing measures taken to ensure the reliability and validity of pain and knee flexion assessments would strengthen the methodological rigor of the study.

Response: Thank you very much for your valuable comment,.We have added a clarification stating that a TSK score greater than 37 is considered to have kinesiophobia. In addition, we elaborate on how we ensure the accuracy of measuring pain and knee flexion.Line 199-206，210-216，221-225，232-236 were added to explain how we ensure accuracy.

Line 199-206:“Prior to the questionnaire survey, we conducted a unified training session for the two team members responsible for the assessment. This training ensured that they were familiar with the content of the questionnaire and the survey process, enabling them to accurately address any questions posed by the patients. During the investigation, the evaluators were instructed to use standardized guidelines to explain the study's purpose and the procedure for completing the questionnaire to the patients.

Line210-216:After the patient completes the questionnaire, the evaluator should conduct a preliminary review of the responses on-site to identify any missing items, obvious errors, or illogical answers, and confirm these with the respondent promptly. Individuals who score greater than 37 points on the questionnaire are classified as having kinesiophobia, while those who score 37 points or less are classified as not having kinesiophobia.

Line 221-225:To ensure accurate pain assessment, the evaluator should provide consistent instructions to explain the Numerical Rating Scale to the patient prior to each evaluation. The assessment should be conducted while the patient is conscious, and the evaluator should confirm the findings with the patient following the assessment.

Line 232-236:To ensure the accuracy of the measurements, each assessment is conducted by the attending physician, who employs standardized instructions to explain the measurement method to the patient. Additionally, the same ruler is utilized to measure the patient's active knee flexion. 

4.comment:

The results section comprehensively presents the findings but could be enhanced by providing more detailed statistical analyses and graphical representations. Specifically, elaborating on the statistical tests used to compare pain scores and knee flexion between groups at different time points would clarify the robustness of the findings. Tables summarizing demographic characteristics and outcome measures should include statistical significance indicators prominently to facilitate interpretation. Moreover, integrating specific patient anecdotes or quotes illustrating the impact of kinesiophobia on pain experiences could enrich the narrative and humanize the findings.

 Response: Due to our limited statistical knowledge, our modifications may not fully meet your expectations. However, we have made minor adjustments within our capabilities to demonstrate the methods employed for tests of normality and homogeneity of variances. We have also modified the last column in the demographic table to reflect statistical differences. Your suggestion to integrate specific patient anecdotes or quotes that illustrate the impact of kinesiophobia on pain experiences is excellent and could indeed enrich the narrative and humanize the findings. Unfortunately, we do not have data on this matter, so we are unable to include patient anecdotes and quotes. We will actively incorporate your suggestions in the next phase of our work.

5.Comments

The discussion effectively contextualizes the study's findings within the existing literature on postoperative pain management and rehabilitation following TKA. It correctly emphasizes the role of psychological factors, such as kinesiophobia, in influencing pain perception and functional outcomes. To further strengthen the discussion, drawing explicit connections between study results and theoretical frameworks, such as fear-avoidance models or biopsychosocial perspectives, would deepen the analysis. Addressing potential confounding variables, such as BMI or preoperative pain severity, in influencing study outcomes would provide a more nuanced interpretation of the findings. Furthermore, discussing implications for clinical practice, such as tailored rehabilitation strategies for patients with high kinesiophobia scores, would enhance the relevance of the study's findings.

Response:Following your suggestion, we have incorporated a discussion of the relationship between the fear-avoidance model and the research results in the Discussion section to enhance the analysis. As you noted, addressing potential confounding variables indeed increases the robustness of the results, and factors such as BMI and preoperative pain may influence the differences observed. In this study, since BMI and preoperative pain did not show differences between the two groups, we did not analyze these variables; however, your suggestion is well-taken. Additionally, we have included relevant literature to enrich the discussion and guide clinical practice.Line333-338,352-361,448-462 were added to To enrich the content of the discussion.

Line 333-338:“The fear-avoidance model is a psychological framework that explains how fear of pain can lead to avoidance behaviors, which in turn can significantly impact a patient's psychological state and recovery process. This model posits that when individuals experience pain, they may develop negative beliefs about physical activity and their ability to cope with pain, leading to a cycle of fear and avoidance[12].”

Line 352-361:”The psychological state of patients affected by the fear-avoidance model can deteriorate as a result of these avoidance behaviors. Patients may experience increased anxiety, depression, and a sense of helplessness due to their inability to engage in normal activities. This emotional distress can further exacerbate their perception of pain and disability, creating a vicious cycle where fear leads to avoidance, which in turn leads to increased pain and disability 4. The model highlights that fear-avoidance beliefs can lead to chronic pain conditions, as the lack of movement and activity can result in physical deconditioning and a reduced ability to cope with pain[14].”

Line448-462:“Assessing the degree of kinesiophobia in patients after knee replacement surgery is crucial for developing personalized intervention plans that can enhance recovery outcomes.Conducting structured interviews can help clinicians understand the patient's experiences and fears related to movement. Open-ended questions about their concerns regarding pain during rehabilitation activities can reveal underlying kinesiophobia[24].Providing patients with clear, consistent information about the recovery process and the importance of movement can help alleviate fears. Educational interventions should focus on the benefits of early mobilization and the expected pain levels during rehabilitation. [7].Encouraging the involvement of family and friends in the rehabilitation process can provide emotional support and reduce feelings of isolation. Research indicates that social support plays a crucial role in recovery, as patients who feel supported are more likely to engage in rehabilitation activities and adhere to exercise regimens[25].”

6.comment

Conclusion:The conclusion succinctly summarizes the study's main findings but could expand to highlight the broader implications for clinical practice and future research directions. Emphasizing the need for integrated interdisciplinary approaches involving physical therapy, psychology, and anesthesia in managing patients with kinesiophobia post-TKA would underscore the study's clinical relevance. Additionally, outlining specific recommendations for enhancing patient education and support to mitigate kinesiophobia-related barriers to rehabilitation would provide actionable insights for healthcare providers.

Response:Based on your suggestions, we have added discussions on integrating multidisciplinary teams and the significance of improving social support in the conclusion section to enrich our conclusions(Line 480-489).

Line480-489: A comprehensive, multidisciplinary approach that includes physiotherapists, psychologists, and occupational therapists can enhance recovery. This team can collaboratively address both the physical and psychological aspects of rehabilitation, ensuring that patients receive holistic care that targets kinesiophobia alongside physical recovery[26]. Social support plays a crucial role in recovery, as patients who feel supported are more likely to engage in rehabilitation activities and adhere to exercise regimens[27].In short, early identification of patients with kinesiophobia after TKA and timely intervention are necessary and beneficial.

Special thanks to you for your good comments.

Reviewer #2:

1.comment

after the revision the article improved significantly and can be considered for publication in the journal

Response:Thank you very much for the affirmation of our work.

Editor:

1.please revise the article according to the reviewers' suggestions and submit it for final decision. In addition, please submit an English copy of the ethical approval report.

Response:Thank you, we have now been through and done this.

2.Please review your reference list to ensure that it is complete and correct. If you have cited papers that have been retracted, please include the rationale for doing so in the manuscript text, or remove these references and replace them with relevant current referenc

---

## [Editor Report · Decision Letter 2]

23 Dec 2024

PONE-D-24-30449R2Study on the impact of kinesiophobia after Total knee Arthroplasty on the rehabilitation of patients during hospitalizationPLOS ONE

Dear Dr. Yao,

Thank you for submitting your manuscript to PLOS ONE. After careful consideration, we feel that it has merit but does not fully meet PLOS ONE’s publication criteria as it currently stands. Therefore, we invite you to submit a revised version of the manuscript that addresses the points raised during the review process.

Please submit your revised manuscript by Feb 06 2025 11:59PM. If you will need more time than this to complete your revisions, please reply to this message or contact the journal office at plosone@plos.org. Please include the following items when submitting your revised manuscript:A rebuttal letter that responds to each point raised by the academic editor and reviewer(s). You should upload this letter as a separate file labeled 'Response to Reviewers'.A marked-up copy of your manuscript that highlights changes made to the original version. You should upload this as a separate file labeled 'Revised Manuscript with Track Changes'.An unmarked version of your revised paper without tracked changes. You should upload this as a separate file labeled 'Manuscript'.If applicable, we recommend that you deposit your laboratory protocols in protocols.io to enhance the reproducibility of your results. Protocols.io assigns your protocol its own identifier (DOI) so that it can be cited independently in the future. For instructions see: https://journals.plos.org/plosone/s/submission-guidelines#loc-laboratory-protocols. Additionally, PLOS ONE offers an option for publishing peer-reviewed Lab Protocol articles, which describe protocols hosted on protocols.io. Read more information on sharing protocols at https://plos.org/protocols?utm_medium=editorial-email&utm_source=authorletters&utm_campaign=protocols.

We look forward to receiving your revised manuscript.

Kind regards,

André Pontes-Silva

Academic Editor

PLOS ONE

Journal Requirements:

**Additional Editor Comments:**

Dear authors, thank you for submitting your article in accordance with the requests of the editor and reviewers. After careful reading, I would like to request some adjustments for the final decision.

—Include the study design in the title;

—Include keywords according to MeSH terms (https://www.ncbi.nlm.nih.gov/mesh/);

—The objective stated in your abstract is different from the objective stated in your introduction. Please correct this. The sentence describing the objective must be exactly the same in both;

—In the last paragraph of the introduction, state a hypothesis;

—Adjust the methods into topics according to the study design (https://www.equator-network.org/). In addition, in the methods section, include a topic called "Sample Size" and explain in detail the sampling strategies used in this study. The statistical analysis part should not include sampling;

—Include the conclusion as a short-topic (describe it in 1 paragraph). The conclusion described in the abstract and the conclusion described after the discussion must be exactly the same.

---

## [Author Response · Author response to Decision Letter 2]

24 Dec 2024

Dear Editors and Reviewers:

Thank you for your letter and for the reviewers’ comments concerning our manuscript entitled “Study;the impact of Kinesiophobia after Total Knee Arthroplasty on the rehabilitation of patients during hospitalization” (ID:PONE-D-24-30449R1). Those comments are all valuable and very helpful for revising and improving our paper, as well as the important guiding significance to our researches. We have studied comments carefully and have made correction which we hope meet with approval. Revised portion are marked highlight in the paper. The main corrections in the manuscript and the responds to the reviewer’s comments are as flowing: Responds to the reviewer’s comments:

1.comment：Include the study design in the title;

Response:Thank you very much for your valuable comment.Given our sample size and method of subject selection, we classify our study as “A Pilot Study”.This is our revised title.”Study on the impact of Kinesiophobia after Total Knee Arthroplasty on the rehabilitation of patients during hospitalization：A Pilot Study

”

2.comment: Include keywords according to MeSH terms

Response:Thank you very much for your valuable comment.We have modified our keywords in accordance with the requirements of meSH terms.This is our modified keyword.”Kinesiophobia Total Knee Arthroplasty Rehabilitation

“

3.comment: The objective stated in your abstract is different from the objective stated in your introduction. Please correct this. The sentence describing the objective must be exactly the same in both;

Response:Thank you very much for your valuable comment.We have revised the research objectives in the abstract and introduction to ensure they are fully consistent.

4.comment:In the last paragraph of the introduction, state a hypothesis;

Response:Thank you very much for your valuable comment.As you requested, we state our hypotheses in the last paragraph of the introduction.The following is the hypothesis section we added：

“Based on the preceding discussion and existing research, we propose the following hypotheses. Hypothesis 1: At various measurement time points post-surgery, patients with kinesiophobia will exhibit higher resting pain scores than those without kinesiophobia. Hypothesis 2: At various measurement time points post-surgery, patients with kinesiophobia will demonstrate poorer knee joint flexion compared to those without kinesiophobia. Hypothesis 3: Compared to patients without kinesiophobia, those with kinesiophobia will experience a delayed time to ambulate for the first time after surgery, engage in shorter durations and distances of activity, and report higher pain scores during their initial mobilization. Furthermore, patients with kinesiophobia are expected to have longer postoperative hospitalization durations and perform worse on the two-minute walk test on the day of discharge.

”

5.comment:Adjust the methods into topics according to the study design (https://www.equator-network.org/). In addition, in the methods section, include a topic called "Sample Size" and explain in detail the sampling strategies used in this study. The statistical analysis part should not include sampling;

Response:Thank you very much for your valuable comment.At your request, we have added a topic “Sample Size” to the Methods section and, in the first paragraph of the research process, elaborated on how we sampled our study subjects.The following is the part we added:

“Sample size

Sample size calculation was performed using power analysis and sample size (PASS) software (version 2021).The pain scores of the research object were used as the observed outcome indicators. The value of alpha was set as 0.05, power was set as 0.9, number of repeated measurements set as 6, and according to previous literature[9], means was set as 3, standard deviation was set as 3, conditional correlation coefficient set as 0.5.The result showed that one group was expected to recruit 31 patients.” 

“Research process

This study has been approved by the Ethics Committee of Baoding No.1 Central Hospital. Informed consent forms were signed by all participating patients. In the sampling process of this study, given the constraints of limited research resources and time, we employed a convenience sampling strategy. Specifically, during the study period, investigators assessed whether patients admitted to the hospital met the inclusion criteria by reviewing the medical record system. If a patient satisfied the inclusion criteria, the investigator would bring the informed consent form to the patient’s bedside, provide a brief summary of the study's objectives, and explain the detailed research process. The patient would also be informed of their right to withdraw from the study at any time. Should the patient agree to participate, they would be required to sign the informed consent form. Conversely, if a patient declined to participate, they were assured that this decision would not negatively affect their treatment during hospitalization.

”

6.comments:Include the conclusion as a short-topic (describe it in 1 paragraph). The conclusion described in the abstract and the conclusion described after the discussion must be exactly the same.

Response:Thank you very much for your valuable comment.We have streamlined our conclusion sections and made sure they are consistent across the conclusion and abstract sections.The following is our revised conclusion section:

“Conclusion

In short,compared to patients without kinesiophobia, the presence of kinesiophobia after TKA surgery significantly impacts the efficacy of early rehabilitation exercises during hospitalization. This impact is observed in pain perception, knee joint mobility, the 2-minute walk test ,etc. Early identification of patients with kinesiophobia after TKA and timely intervention are necessary and beneficial.”

We tried our best to improve the manuscript and made some changes in the manuscript. These changes will not influence the content and framework of the paper. We appreciate for Editors/Reviewers’ warm work earnestly, and hope that the correction will meet with approval.

Once again, thank you very much for your comments and suggestions.We look forward to hearing from you as soon as possible.

Kind regards

Yichao Yao 

24/12/2024

---

## [Editor Report · Decision Letter 3]

5 Jan 2025

Study on the impact of Kinesiophobia after Total Knee Arthroplasty on the rehabilitation of patients during hospitalization：A Pilot Study

PONE-D-24-30449R3

Dear Dr. Yao,

We’re pleased to inform you that your manuscript has been judged scientifically suitable for publication and will be formally accepted for publication once it meets all outstanding technical requirements.

Kind regards,

André Pontes-Silva

Academic Editor

PLOS ONE
---

## [Editor Report · Acceptance letter]

21 Jan 2025

PONE-D-24-30449R3 

PLOS ONE

Dear Dr. Yao, 

I'm pleased to inform you that your manuscript has been deemed suitable for publication in PLOS ONE. Congratulations! Your manuscript is now being handed over to our production team.

Kind regards, 

on behalf of

Professor André Pontes-Silva 

Academic Editor

PLOS ONE